# Layer 4 of mouse neocortex differs in cell types and circuit organization between sensory areas

Federico Scala[1,2,9], Dmitry Kobak [3,9], Shen Shan [1,2], Yves Bernaerts[3], Sophie Laturnus[3], Cathryn Rene Cadwell [4], Leonard Hartmanis[5], Emmanouil Froudarakis [1,2], Jesus Ramon Castro[1,2], Zheng Huan Tan[1,2], Stelios Papadopoulos[1,2], Saumil Surendra Patel[1,2], Rickard Sandberg [5], Philipp Berens [3,6], Xiaolong Jiang [1,2,7] & Andreas Savas Tolias [1,2,8]

Layer 4 (L4) of mammalian neocortex plays a crucial role in cortical information processing, yet a complete census of its cell types and connectivity remains elusive. Using whole-cell recordings with morphological recovery, we identified one major excitatory and seven inhibitory types of neurons in L4 of adult mouse visual cortex (V1). Nearly all excitatory neurons were pyramidal and all somatostatin-positive (SOM+) non-fast-spiking interneurons were Martinotti cells. In contrast, in somatosensory cortex (S1), excitatory neurons were mostly stellate and SOM+ interneurons were non-Martinotti. These morphologically distinct SOM+ interneurons corresponded to different transcriptomic cell types and were differentially integrated into the local circuit with only S1 neurons receiving local excitatory input. We propose that cell type specific circuit motifs, such as the Martinotti/pyramidal and non-Martinotti/stellate pairs, are used across the cortex as building blocks to assemble cortical circuits.

[1] Center for Neuroscience and Artificial Intelligence, Baylor College of Medicine, Houston, TX, USA. [2] Department of Neuroscience, Baylor College of Medicine, Houston, TX, USA. [3] Institute for Ophthalmic Research, University of Tübingen, Tübingen, Germany. [4] Department of Anatomic Pathology, University of California San Francisco, San Francisco, CA, USA. [5] Department of Cell and Molecular Biology, Karolinska Institutet, Stockholm, Sweden. [6] Department of Computer Science, University of Tübingen, Tübingen, Germany. [7] Jan and Dan Duncan Neurological Research Institute at Texas Children's Hospital, Houston, TX, USA. [8] Department of Electrical and Computational Engineering, Rice University, Houston, TX, USA. [9] These authors contributed equally: Federico Scala, Dmitry Kobak. Correspondence and requests for materials should be addressed to X.J. (email: xiaolonj@bcm.edu) or to A.S.T. (email: astolias@bcm.edu)

The mammalian sensory neocortex is organized as a six-layer structure. In this stereotypical architecture, layer 4 (L4) is the main direct target of sensory inputs coming from the thalamus, thus acting as the first level of cortical processing for sensory signals[1]. Understanding how L4 implements its computations requires to characterize the diversity of its constituent neuronal components and the connectivity among them.

Most previous studies of L4 have focused on primary somatosensory cortex (S1) of young rats and mice. Spiny stellate cells have been reported to be the dominant excitatory cell type, both in rat[2–6] and in mouse[7] (as a result of sculpting of initially pyramidal neurons during development[8,9]). In contrast, inhibitory interneurons are highly diverse in terms of their genetic markers, morphologies and electrophysiological properties[10]. Previous studies have reported three types of fast-spiking (FS), parvalbumin-positive (PV[+]) interneurons[11] and five types of non-FS interneurons[12], all of which have distinct morphologies. Several recent studies revealed that the somatostatin-positive (SOM[+]) interneurons form a single morphological population that has been called non-Martinotti cells[13] since their axons mainly target L4[14,15], in contrast to typical Martinotti cells, which target L1. GABAergic interneuron types exhibit cell-type-specific connectivity patterns. For example, PV[+] FS interneurons receive strong thalamic inputs[16–20] while SOM[+] non-FS interneurons receive weaker thalamic inputs[21,22]. Both groups are reciprocally connected to local excitatory neurons and between each other[11,15,17,19,23], but PV[+] interneurons inhibit each other while SOM[+] interneurons do not[24].

Since most of these studies were performed in S1 of young animals, it is unclear whether the cellular components of L4 and their connectivity profile are the same in adult animals and in other cortical areas. Recent large-scale studies of transcriptomic cell types in mouse and human cortex showed that most interneuron types are shared between cortical areas while the excitatory types are predominantly area-specific[25,26]. In line with this, it has been shown that excitatory cells in L4 of mouse and rat primary visual cortex (V1) are pyramidal[27,28], in contrast to L4 in S1. However, there has been no systematic comparisons of anatomical and electrophysiological properties as well as connectivity profiles between L4 of different cortical areas, leaving an open question about the similarity in their cellular components and circuitry.

To address this question, we compare the microcircuit organization in L4 of V1 and S1 in adult mouse. We perform a thorough census of the morphological cell types in V1 L4 of adult mice (median age 71 days) using multi-cell simultaneous whole-cell recordings combined with post-hoc morphological recovery[29]. We identify several key differences in the cellular composition of L4 in V1 compared to the previous literature on S1, which we verify using targeted recordings of certain cell types in S1 L4 of similarly-aged mice. We further investigate the local connectivity profiles in L4 of both V1 and S1, finding similarities as well as some important differences in their circuitry. In addition, we map SOM[+] cell types in both areas to a reference transcriptomic cell-type atlas[25] using Patch-seq[30–32]. Our findings suggest that the same transcriptomic type can have different morphological and electrophysiological phenotypes depending on the cortical area, possibly adapting to specific neuronal circuits depending on the activity and the environment.

## Results
**Morphological cell types in L4 of adult mouse visual cortex.** We characterized the electrophysiological and morphological features of L4 neurons in V1 of adult mice (n = 129, median age 71 days,

interquartile range 62–85 days, full range 55–330 days, Supplementary Fig. 1) using whole-cell patch-clamp recordings combined with morphological recovery (see Methods). Altogether, we recovered and analyzed the morphology of n = 1174 neurons (578 excitatory, 596 inhibitory).

Out of the 578 excitatory cells, 573 (99.1%) were pyramidal neurons (PYR), characterized by apical dendrites extending into layer 1 (L1), consistent with previous reports in rats[27] and young mice[28]. These neurons did not show a complex dendritic arborization in L1, differing from typical layer 5 (L5) pyramidal neurons, which generally have a prominent tuft in L1[33–35] (Fig. 1a). Only five (0.9%) of the excitatory neurons were classified as spiny stellate cells based on the absence of the apical dendrites extending out of L4 to supragranular layers, and on their symmetrical non-polarized dendritic structure[6]. The prevalence of PYRs among excitatory neurons in L4 of V1 was further supported by the fact that all labeled neurons recorded in Scnn1a-Cre/Ai9 mice (n = 5), a transgenic mouse line that preferentially labels excitatory neurons in L4[36,37], were morphologically confirmed as PYRs (100%, 30/30). In terms of electrophysiology, PYRs exhibited large action potential (AP) width, high AP amplitude, and shallow afterhyperpolarization (AHP), which clearly discriminated them from GABAergic interneurons (Fig. 1b).

Interneurons showed a greater variability in both morphological and electrophysiological features. To facilitate targeting of interneurons for whole-cell recordings, we used Viaat-Cre/Ai9 mice (n = 47)[29,38]. Almost all labeled L4 neurons recorded from these mice (95.5%, 234/245) were morphologically confirmed as interneurons. On the other hand, all unlabeled L4 neurons (n = 133) were morphologically confirmed as excitatory neurons, suggesting that the entire population of interneurons in L4 was labeled in this Cre line. We identified seven GABAergic cell types (Fig. 1a) based on their morphology, following a widely used classification scheme based on the axonal geometry and projection patterns[29,39–41]. We used several other Cre lines (PV-Cre/Ai9, n = 31; SOM-Cre/Ai9, n = 14; VIP-Cre/Ai9, n = 8) to relate genetic markers with morphological cell types (Supplementary Fig. 2).

We found four distinct types of PV[+], fast-spiking (FS) cells with narrow AP width, high maximal firing rate, and thick axons. None of them had chandelier-like axonal structures, so we assumed that all four types were basket cells with different axonal morphologies[10]: the most common were large basket cells (LBCs; 37.6% among the interneurons labeled in Viaat-Cre, 88/234), followed by small basket cells (SBCs; 9.4%, 22/234), double-bouquet basket cells (DBCs; 5.6%, 13/234), and horizontally elongated basket cells (HBCs; 3.4%, 8/234). Here we adopted the terminology similar to the one we used previously for L2/3 and L5 interneurons[29] and in particular define double-bouquet cells as having two wide axonal bundles of similar size going in opposite directions.

Martinotti cells (MCs; 20.1%, 47/234), characterized by an ascending axon that projected to L1 and by the large membrane time constant, were the only SOM[+] cell type. Previous literature suggested that Martinotti cells in L5 form two groups with different morphology and/or electrophysiology[29,42], but we did not observe such heterogeneity in L4: in particular, none of the MCs showed a rebound firing after stimulation offset[42]. A small fraction of SOM[+] neurons exhibited an FS firing pattern and their morphological features matched those of LBCs (8.2%, 5/61; Supplementary Fig. 2), in agreement with the previous reports that due to potential off-target recombination, ~5–20% of neurons labeled in SOM-Cre line are FS[29,43,44] and PV[+]/SOM[−] at the protein level[43].

VIP[+] neurons were represented by the bipolar cells (BPCs; 12.4%, 29/234) with a small soma, dendrites extending to L1

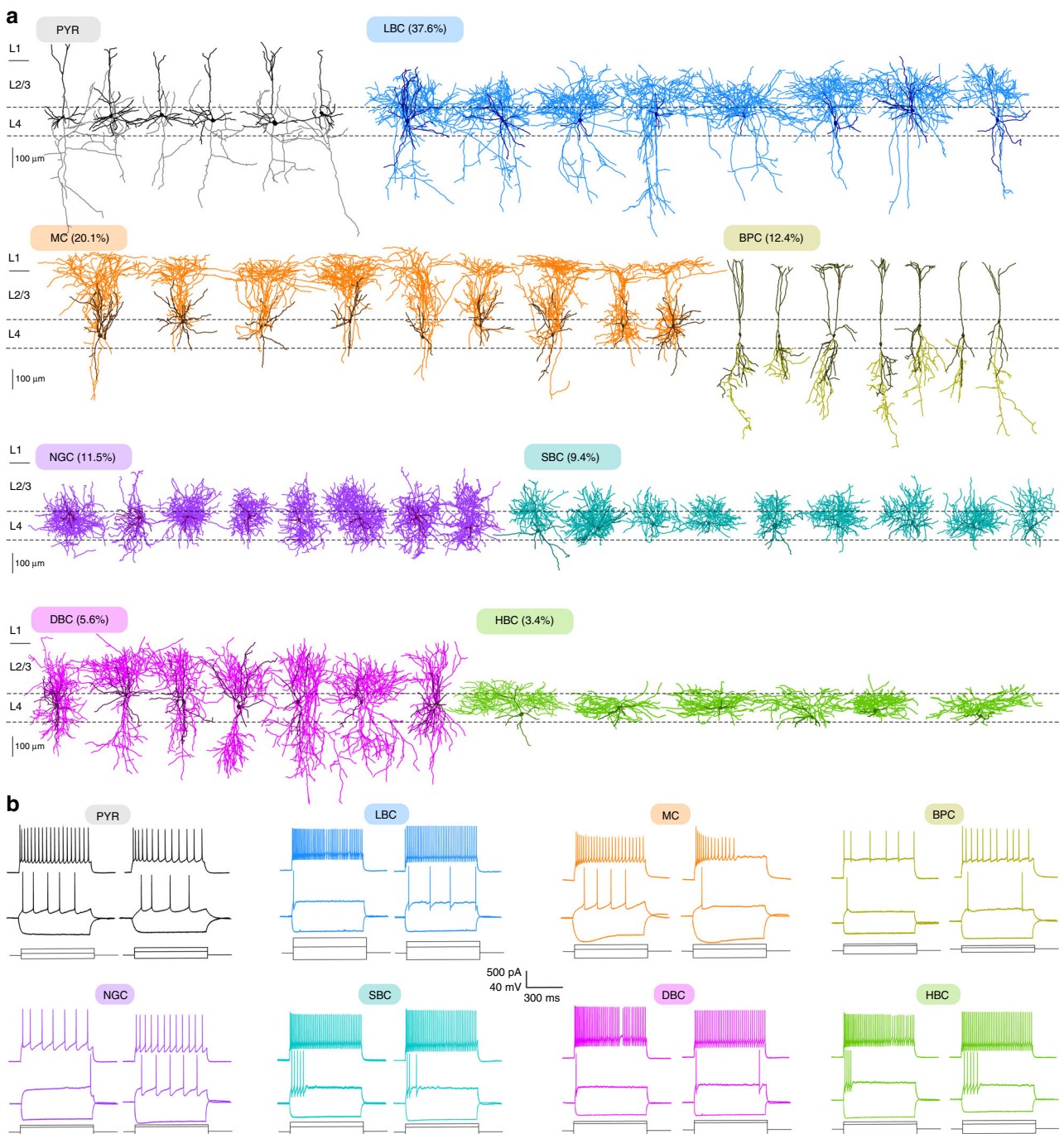

**Fig. 1** Morphological cell types in V1 L4. **a** Representative morphologies for each cell type. The dendrites are shown in a darker shade of color and axons in a lighter shade. Types are sorted by abundance from high to low. Fractions indicate the proportions among the neurons labeled in Viaat-Cre mice. PYR: pyramidal cells, LBC: large basket cells, MC: Martinotti cells, BPC: bipolar cells, NGC: neurogliaform cells, SBC: small basket cells, DBC: double-bouquet basket cells, HBC: horizontally elongated basket cells. **b** Spiking responses to step currents for two exemplary cells from each of the eight morphologically defined cell types

and L5, an axon projecting mostly downward to L5, and an irregular firing pattern. The last type were neurogliaform cells (NGCs; 11.5%, 27/234), characterized by a very thin axon highly ramifying around the soma and a late-spiking firing pattern with a large AP width. We did not encounter any labeled NGCs in PV-Cre, SST-Cre, or VIP-Cre lines, suggesting that they belonged to the *Lamp5* transcriptomic family[25]. A more detailed description of morphological and electrophysiological properties of all interneuron types can be found in the Supplementary Note 1.

To support our expert classification, we fully reconstructed a subset of neurons from each inhibitory type ($n = 92$ in total) and trained a regularized logistic regression classifier to discriminate between each pair of inhibitory cell types (see Methods). We used 2D density maps and a set of morphometric statistics (Supplementary Fig. 3) as predictors[45]. Across all 21 pairs, the average cross-validated classification accuracy was 0.92, with most pairs discriminated almost perfectly (Fig. 2a, left). However, SBC/HBC pair showed only ~0.6 classification accuracy. Visualization of this dataset with t-SNE (Fig. 2a, right) indicated that SBC/HBC

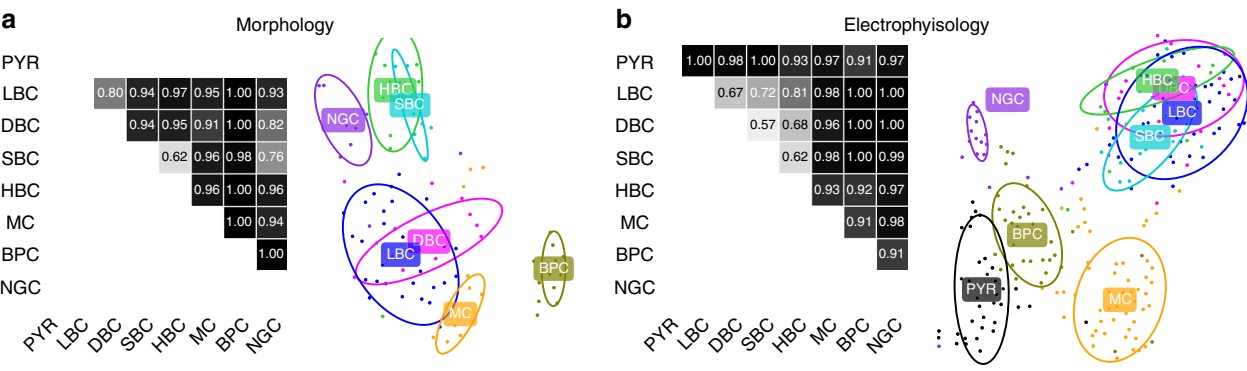

**Fig. 2** Discriminability of cell types in V1 L4. **a** Cross-validated pairwise classification accuracy for each pair of inhibitory cell types, using regularized logistic regression on a diverse set of morphological features. Total sample size $n = 92$. Right: 2D visualization of the same $n = 92$ cells in the space of morphological features using t-SNE. Ellipses show 80% coverage assuming 2D Gaussian distributions and using robust estimates of the mean and the covariance (i.e., ellipses do not include outliers). **b** Cross-validated pairwise classification accuracy for each pair of cell types, using electrophysiological features. Total sample size $n = 235$. Right: 2D visualization of the same $n = 235$ cells in the space of electrophysiological features using t-SNE

types were partially overlapping, as well as LBC/DBC. Overall, this analysis suggests that while most morphological classes can be very well discriminated, some may be partially overlapping. An important caveat is that low classification accuracy can also be due to an insufficient sample size.

To further explore variability in electrophysiological properties between morphological cell types, we characterized the firing pattern and intrinsic membrane properties of a subset of neurons ($n = 235$) using 13 automatically extracted electrophysiological features (Supplementary Fig. 4). Most features exhibited strong differences between cell types (Supplementary Fig. 5). Two-dimensional visualization of this dataset using t-SNE (Fig. 2b) showed that all four PV$^+$ cell types overlapped in one group of electrophysiologically similar FS neurons, while the other four types (PYR, NGC, BPC, and MC) each had distinct firing patterns. We confirmed this using pairwise classification with regularized logistic regression (Fig. 2b): the average cross-validated classification accuracy between the FS types was only 0.67, while the average accuracy across all other pairs was 0.98.

**V1 differs from S1 in major L4 cell types**. In contrast to L4 V1, spiny stellate cells are known to be abundant in S1 L4 of rats and mice[2–8]. To confirm this, we recovered L4 excitatory cells in S1 ($n = 24$ mice, including $n = 5$ Scnna1-Cre/Ai9) with the same method as in V1. We found that indeed 76.6% (85/111) of the recovered spiny neurons did not have a clear apical dendrite and were thus classified as spiny stellate cells (Fig. 3b), while the remaining 23.4% were pyramidal cells. This confirms that, unlike in V1, spiny stellate cells are the predominant excitatory population in L4 of adult mouse S1 (Fisher's exact test for difference in the fraction of spiny stellate cells between V1 and S1: $p < 0.0001$; 85/111 vs. 5/578).

Recent evidence indicates that most, if not all, L4 SOM$^+$ cells in mouse S1 have axons mostly localized within L4, in stark contrast to typical MCs[15], and have been called non-Martinotti cells (NMCs)[13]. Indeed, we found that in S1, almost all L4 SOM$^+$ neurons we recovered (96.2%, 76/79, from $n = 19$ SOM-Cre/Ai9 mice) had NMC morphology characterized by a thin, highly ramifying axon mostly residing within L4 (Fig. 3b). Only two cells showed an ascending axon projecting to L1 typical of MCs (2.5%, 2/79) and one was characterized by a thick axon branching similarly to LBC with a FS firing pattern (1.3%, 1/79) (Fisher's exact test for difference in the fraction of NMCs between S1 and V1: $p < 0.0001$; 76/79 vs. 0/61).

The NMCs also differed in their firing pattern from MCs recorded in V1: they had a higher maximal firing rate, a lower AP width, and a lower membrane time constant (Fig. 4a, Supplementary Fig. 5). This resembles the FS firing pattern, and one previous study called NMCs "quasi-FS"[14]. Comparison of electrophysiological features between MCs, NMCs, and FS cells revealed that NMCs were in between MCs and FS cells in terms of their firing patterns and intrinsic membrane properties (Supplementary Figs. 5, 6).

To further investigate the differences between MCs in V1 and NMCs in S1, we used the Patch-seq[30–32] technique, which combines patch-clamp recordings with single cell transcriptomics. Using $n = 6$ SOM-Cre/Ai9 mice, we sequenced the RNA of labeled neurons in L4 of V1 and S1 ($n = 42$ in V1 and $n = 35$ in S1 after quality control), and also in L5 of each area ($n = 17$ and $n = 16$ respectively). We obtained on average 1.1 million reads per cell (median; mean ± SD on a log10 scale: 6.0 ± 0.3) and detected 6.4 ± 1.6 thousand (mean ± SD) genes per cell (Supplementary Fig. 7). We mapped these cells to a large transcriptomic cell-type dataset[25] that contained 21 somatostatin cell types with 2880 neurons from V1 and anterior lateral motor cortex (ALM). The quality of the mapping was equally good for V1 and S1 cells (Supplementary Fig. 7), suggesting that the V1 + ALM dataset is a reasonable reference for S1 interneurons, in agreement with the idea that inhibitory transcriptomic cell types are shared across cortical regions[25,26]. Three cells (excluded from the cell counts given above and from further analysis) had fast-spiking firing pattern and mapped to *Pvalb Reln Itm2a* transcriptomic type, likely corresponding to the basket cells that we found labeled in the SOM-Cre line (Fig. 3). These three cells did not express SOM (zero read count), suggesting that they could have transiently expressed it during development, as hypothesized by Hu et al.[43] All other cells mapped to *Sst* transcriptomic types.

Most L4 cells (81%, 62/77) were assigned to *Sst Calb2 Pdlim* and *Sst Hpse Cbln4* transcriptomic types (Fig. 4b, e), with S1 cells falling almost exclusively into the *Hpse* type (27/29) and V1 cells falling preferentially into the *Calb2* type (21/33) ($p < 0.0001$, Fisher's exact test). This suggests that *Sst Calb2 Pdlim* is a MC type, in agreement with the conclusions of Tasic et al.[25] based on the data from Paul et al.[46], and that *Sst Hpse Cbln4* is a NMC type, in agreement with Naka et al.[47]. However, this raises the question of why some V1 L4 SOM$^+$ cells, none of which had a NMC morphology (see above), had a NMC transcriptomic profile, both among our Patch-seq cells and in the Tasic et al. dataset[25].

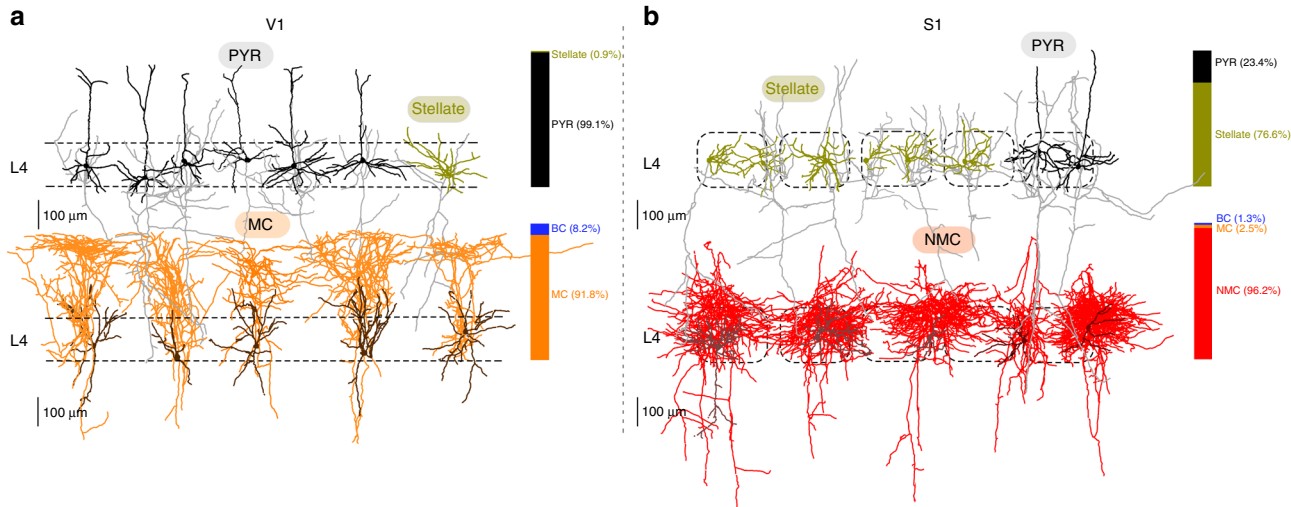

**Fig. 3** V1 differs from S1 in excitatory cells and SOM+ interneurons in L4. **a** Representative morphologies of excitatory and SOM+ neurons in V1 L4. Bar graphs indicate the fractions of each cell type among all morphologically recovered excitatory neurons (top) and all morphologically recovered neurons labeled in the SOM-Cre line (bottom). **b** The same in S1 L4. Dashed rectangles represent individual cortical barrels

To answer this question, we looked at electrophysiological features that were most different between SOM+ interneurons in V1 and S1 (Cohen's $d > 1$: input resistance, membrane time constant, maximum firing rate, and AP width) and found that for two of them (input resistance and membrane time constant) V1 cells belonging to the *Hpse* type had values more similar to the S1 cells than to the V1 cells from the *Calb2* type (Fig. 4c). This suggests that electrophysiologically, V1 *Hpse* MC cells are in between V1 *Calb2* MC cells and S1 NMC cells.

The relationship between gene expression and electrophysiological features can be visualized using the sparse reduced-rank regression analysis that we have recently introduced[48]. This technique aims to reconstruct all the electrophysiological features using a two-dimensional projection of the expression levels of a small set of genes (Fig. 4d). The optimal number of genes was found using cross-validation (see Methods). This analysis supports our conclusion that V1 *Hpse* MCs are in between *Calb2* MCs and NMCs in terms of electrophysiology. Interestingly, this analysis also showed that some of the cells assigned to the *Tac1* and *Mme* types had a distinct fast-spiking-like firing pattern, which was different from firing patterns of MCs and NMCs (but was not as sustained as the proper FS pattern). These three SOM+ transcriptomic cell types have recently been identified in Tasic et al.[25], and do not have known morphological or electrophysiological counterparts.

The L5 SOM+ cells that we sequenced in both areas mostly mapped to a different set of transcriptomic types than the L4 SOM+ cells, but there were no apparent differences between S1 and V1 in terms of transcriptomic cell types (Fig. 4b). Morphologically, all non-fast-spiking SOM+ cells in V1 L5[29,47] and the majority of SOM+ cells in S1 L5[22,29,42,47] are known to be MCs. In agreement with that, we found that L5 SOM+ cells had electrophysiological features similar to L4 MCs (Fig. 4d).

**Connectivity among excitatory and SOM+ neurons.** So far, we have described major differences in the morphology, electrophysiology, and transcriptomic signatures of excitatory neurons and SOM+ interneurons in L4 between V1 and S1. We next investigated whether there are differences in their connectivity profiles as well, using simultaneous multi-cell patch-clamp recordings. We found that certain connectivity patterns between

them are very similar across both areas (Fig. 5). First, the connection probabilities among excitatory cells were low in both areas (1.0%, 7/701 in V1; 2.5%, 3/122 in S1). Second, the connection probabilities between SOM+ cells were also low in both areas (0%, 0/68 in V1; 3.8%, 2/52 in S1). Third, the connection probabilities from SOM+ cells to excitatory cells were high in both areas (21.1%, 30/142 in V1, 26.6%, 17/64 in S1). In addition, despite their low connectivity via chemical synapses, both MCs in V1 and NMCs in S1 were similarly often interconnected by gap junctions (MCs: 23.5%, 8/34; NMCs: 30.7%, 8/26; Supplementary Fig. 8).

On the other hand, we found a striking area-specific difference in connection probabilities from excitatory to SOM+ neurons. In S1, NMCs received facilitating synaptic connections from local excitatory neurons (12.5%, 8/64), in line with previous studies in young rodents[15,17]. In contrast, we did not find any connections (0%, 0/142) from local excitatory neurons to MCs in V1 ($p = 0.0002$, Fisher's exact text). This was also in stark contrast to MCs in L2/3 and L5 of adult mouse V1, which receive strong facilitating synaptic inputs from local PYRs in the same layers[29] (see Discussion for further considerations).

In addition, we tested the connectivity of LBCs in V1 L4 (Fig. 5b, Supplementary Fig. 10). We found that LBCs followed the same connectivity rules as previously found in other layers[29,49]. PYRs connected to LBCs with probability 12.5% (38/303), MCs inhibited LBCs with probability 32.6% (15/46), and LBCs inhibited each other (36.7%, 75/204), MCs (13.0%, 6/46) and PYRs (25.7%, 78/303). All of these connection patterns have also been reported in S1 L4 of young mice[24]. We also found that LBCs were electrically coupled to each other with probability 27.5% (28/102) but were never electrically coupled to MCs (0/23), in agreement with previous findings that gap junctions exist between inhibitory cells of the same type[50].

Notably, the connection probability between PYRs in V1 L4 was very low, consistent with our previous work in other layers in adult animals[29], but in contrast to the findings in young and juvenile rodents[51,52]. To confirm that this low connectivity reflects an age effect, we measured the connectivity between PYRs in V1 L4 at different ages (P15-20 and P30-40, $n = 5$ each) using Scnn1a-Cre/Ai9 mice. We found that the connection probability monotonically decreased with age (Supplementary Fig. 11): from 13.2% in P15-20 (15/114) to 5.1% in P30-40 (8/156) to 1.0% (7/701) reported above for the P55+ mice with median age P71.

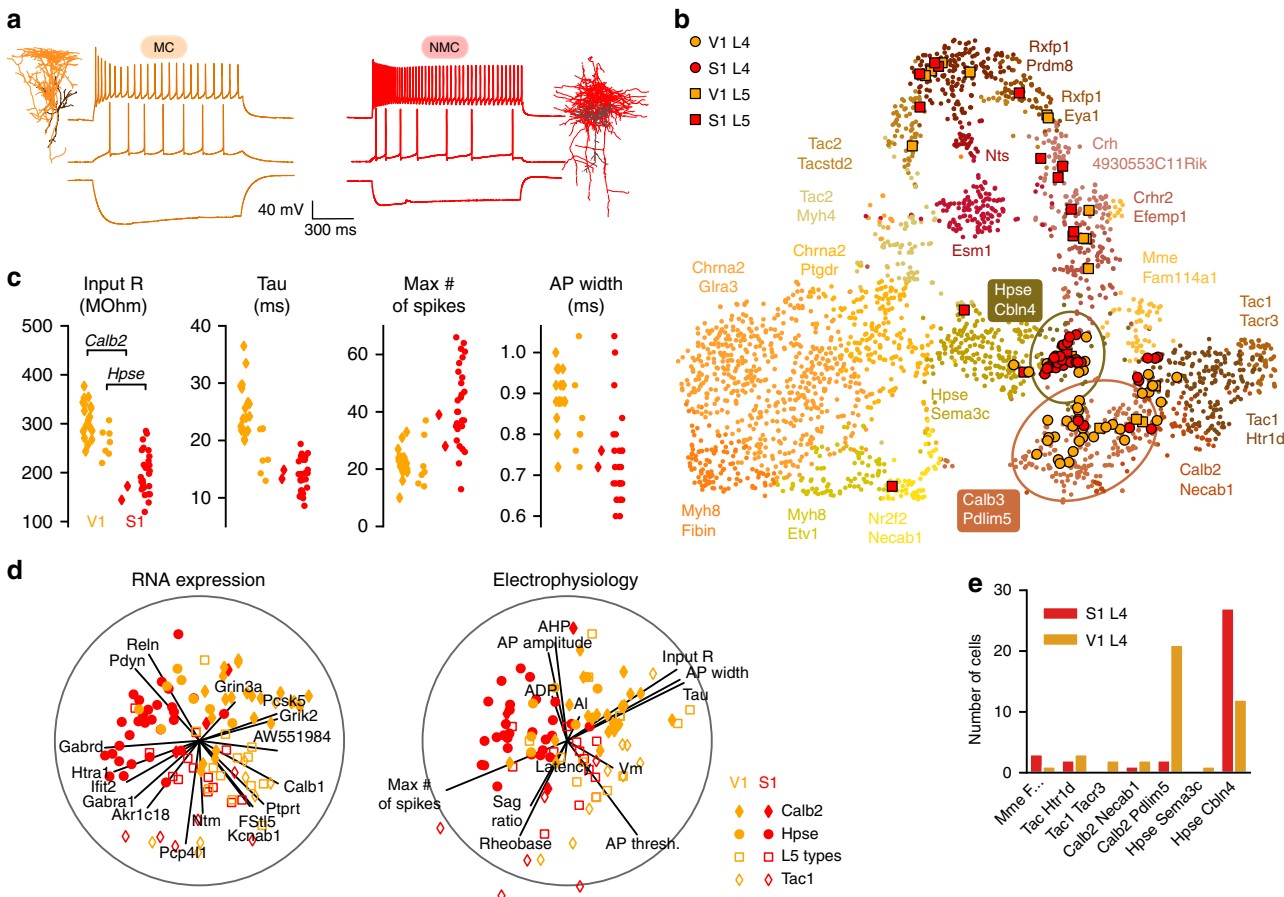

**Fig. 4** Transcriptomic and electrophysiological differences between L4 SOM⁺ interneurons in V1 and S1. **a** Morphologies and firing patterns of two exemplary cells, from V1 (orange) and S1 (red), respectively. **b** Mapping of the Patch-seq cells ($n = 110$) to the t-SNE visualization of the transcriptomic diversity among *Sst* types from Tasic et al.[25] t-SNE was done on all cells from *Sst* types except for *Sst Chodl* that is very well separated from the rest (20 clusters; $n = 2701$ cells), using 500 most variable genes (see Methods). Two ellipses show 90% coverage areas of the two types where the most Patch-seq cells land. Mapping to t-SNE was performed as we described elsewhere[85], see Methods. "Sst" was omitted from type names for brevity. **c** Four electrophysiological features that differed most strongly (Cohen's $d > 1$) between V1 L4 and S1 L4 cells. Only cells assigned to *Sst Calb2 Pdlim5* and *Sst Hpse Cbln4* types are shown. Note that the values are not directly comparable to those shown in Supplementary Fig. 5 because Patch-seq experiments used a different internal solution compared to regular patch-clamp experiments without RNA extraction. **d** Sparse reduced-rank regression analysis[48]: the left biplot shows two-dimensional projection in the transcriptomic space that is optimized to reconstruct the electrophysiological features. The right biplot shows the corresponding two-dimensional projection in the electrophysiological space; it should match to the left plot if the model is accurate. Color denotes brain area (orange for V1, red for S1), marker shape denotes transcriptomic type that each cell was assigned to (circles: *Hpse Cbln4* type; diamonds: *Calb2 Pdlim* type; open diamonds: three *Tac1/Mme* types and the neighbouring *Calb2 Necab1* type; open squares: all other types). Individual electrophysiological features and genes selected by the model are depicted with lines showing their correlations to the two components. Circles show maximal possible correlation. Cross-validated estimate of the overall R-squared was 0.14, and cross-validated estimates of the correlations between the horizontal and vertical components were 0.69 and 0.49, respectively. **e** Type assignments of the Patch-seq cells from L4

This is in agreement with a recent study that found 6.3% (20/315) connection probability in V1 L4 of P46 mice[53].

When measuring connectivity in S1 L4, no special care was taken to ensure that the tested cells were within the same barrel. At the same time, it is known that cells in S1 L4 preferentially make intra-barrel connections[3,4]. To address this concern, we performed a separate series of experiments using $n = 8$ Scnn1a-Cre/Ai9 mice to test intra-barrel connectivity of excitatory neurons. We used the tdTomato fluorescence to detect the barrels during patch-clamp recordings[54] and performed cytochrome oxidase staining in a subset of slices to confirm that the fluorescence signal reliably corresponded to the barrel boundaries[4] (Supplementary Fig. 12). The measured connection probability was 5.2% (5/104), which was larger than the value reported above (2.5%, 3/122) but not significantly different from it ($p = 0.48$, Fisher's exact test). Both estimates are substantially lower than the existing estimates of intra-barrel connectivity

obtained in young animals (30–35%)[3,4,55], which is in line with the decrease in local excitatory connectivity with age discussed above for V1 (Supplementary Fig. 11).

## Discussion

Layer 4 (L4) of mammalian neocortex is the main target of sensory inputs coming from the thalamus. While it is known that its cellular architecture differs between species and between sensory cortical areas[1], here we demonstrated that the difference is not confined to the excitatory neurons, but extends to the somatostatin-positive interneurons and their connectivity.

We described eight morphological cell types in L4 of primary visual cortex (V1) in adult mice as well as the connectivity between the three most abundant cell types. We found that nearly all excitatory neurons in V1 L4 are pyramidal cells (as was previously described in rats[27], guinea pigs[56], and young mice[28]), in

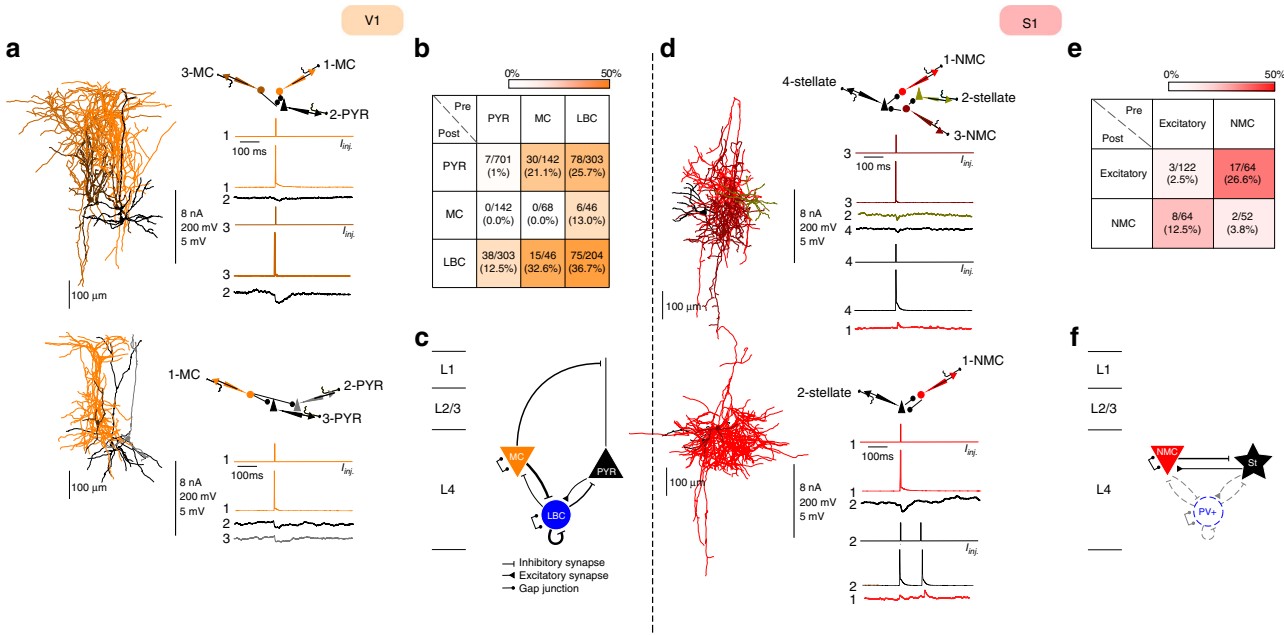

**Fig. 5** Connectivity between excitatory and SOM+ cells in L4 of V1 and S1. **a** Examples of simultaneous recordings from excitatory and SOM+ neurons in V1 L4. Recorded neurons were close to each other (generally less than 150 μm). Vertical scale bar indicates: amplitudes of injected currents in nA, amplitude of APs in mV and amplitude of uEPSPs or uIPSPs in mV. **b** Color-coded connectivity matrix shows the connection probability between cell types as a percentage of tested potential connections. Averages of uEPSPs and uIPSPs as well as PPRs are reported in Supplementary Fig. 9. For the connectivity involving LBCs, see also Supplementary Fig. 10. **c** Schematic of the local circuitry in L4 V1. For gap junctions, see Supplementary Fig. 8. Line thickness corresponds to connection probability. **d–f** The same for L4 S1. In the schematic (**f**), the connectivity rate involving PV+ interneurons (mostly LBCs) is taken from Ma et al.[24]. All the PV+ connections are shown with the same strength as that study used juvenile (P15) mice and so connection strengths are not directly comparable to the values obtained in our experiments. Regarding gap junctions between FS interneurons see also Galaretta et al. and Gibson et al.[50,86]

stark contrast to S1 L4 where the majority of excitatory neurons are spiny stellate cells[6,8,57]. In this work, we did not distinguish between pyramidal and star pyramidal cells[6], and classified excitatory neurons into pyramidal and spiny stellate. Notably, L4 spiny stellate cells in ferret V1 and mouse S1 develop postnatally from neurons that resemble pyramidal cells with an upward projecting apical dendrite[8]. The near absence of spiny stellate cells in V1 L4 of adult mice (as old as 11 months in our experiments) suggests a different developmental path in this case.

Our data indicate that non-fast-spiking SOM+ neurons in V1 L4 are predominantly Martinotti cells, which is in contrast to S1 L4 where almost all SOM+ neurons are non-Martinotti[14,15]. Using Patch-seq, we showed that SOM+ MCs in V1 L4 and SOM+ NMCs in S1 L4 correspond to two different transcriptomic cell types (*Sst Calb2 Pdlim* and *Sst Hpse Cbln4* respectively) previously identified in a large-scale transcriptomic cell atlas[25]. Even though we did not identify any NMCs in V1, the transcriptomic reference dataset[25] contained many V1 cells from the *Sst Hpse Cbln4* type, and we found that around a third of MCs from V1 had transcriptomic profile mapping to this type. These cells show an electrophysiological profile intermediate between MCs and NMCs, but morphologically correspond to MCs based on our data. We hypothesize that these cells may be latent NMCs, present in V1, but failing to develop an NMC morphology due to the nearly complete absence of spiny stellate cells in V1. This example suggests that cells from the same transcriptomic type can exhibit different phenotypes in terms of morphology and electrophysiology, depending on the cortical area. Tasic et al.[25] showed that the majority of transcriptomic inhibitory types are shared between two very different cortical areas (V1 and ALM). Our findings indicate that this does not necessarily imply that morphological types are also all shared.

In terms of connectivity, both MCs in V1 and NMCs in S1 avoided connecting to each other (apart from forming gap junctions), and projected to excitatory population in L4. Moreover, the axonal morphologies of these two cell types seemed to match the respective dendritic morphologies of their excitatory neuronal targets. In V1, axons of L4 MCs primarily projected to L1 where they are potentially able to synapse onto the tuft of L4 PYRs, similar to the pattern described in other cortical layers[58,59]. In S1, by contrast, axons of L4 NMCs were more localized, matching the more compact dendritic structure of spiny stellate cells. This observation is in line with previous findings that the excitatory identity controls the survival and wiring of local interneurons[60,61]. We suggest that the difference in the morphology of SOM+ neurons between these two cortical areas might be a result of the difference in dendritic arborization of the targeted excitatory neurons. Consistent with this, in S1 L5, where the principal excitatory cells are pyramidal and SOM+ interneurons are either L5 MC or L5 NMC, L5 pyramidal cells are preferentially innervated by L5 MCs, not L5 NMCs[47]. We hypothesize that the reshaping of excitatory neurons' apical dendrites in S1 L4 during development, which depends on the sensory input[8], could be followed by the corresponding reshaping of SOM+ neurons. It will be interesting to test whether this MC/pyramidal and NMC/stellate paring exists in other cortical areas and other species.

On the other hand, while we found that SOM+ cells received inputs from local excitatory neurons in S1 L4, in agreement with previous studies[15,24], we did not detect any connections from L4 PYRs to L4 MCs in V1. SOM+ MCs in other layers are known to receive facilitating excitatory inputs from local principal neurons in both S1[62,63] and V1[29,64]. However, our results suggest that L4 MCs in V1 behave differently. Interestingly, previous studies have

shown that in V1, L4 MCs also receive weak inputs from the thalamus compared to other interneuron types[21,22]. Within S1, Naka et al.[47] showed that L4 excitatory neurons connect to NMCs in L5 but not to MCs in L5, which together with our findings, suggests that even across layers, spiny stellate cells do not target MCs but only NMCs. Further investigations are needed to test whether L4 MCs in V1 are driven by PYRs in other layers or by long-range inputs from other areas.

We found very low connection probability between PYRs in V1 L4, which was consistent with the findings in V1 L2/3 and V1 L5 of adult mice[29], but much lower than what was reported in young animals[65,66]. We directly showed that this difference in connection probability among excitatory neurons is due to the age of the animal (Supplementary Fig. 11). One possible caveat here is that our recordings were done at the soma and could have failed to detect synaptic connections formed in the distal dendrites. However, we see no reason to believe that this problem would be more severe in adult animals compared to the young ones (P15–20) and so we do not think that our findings can be an artefact caused by this issue. That said, further investigations would be necessary to obtain a more detailed understanding of neuronal connectivity during development.

It will be important to understand how the difference in circuit organization of L4 between S1 and V1 affects the computations performed in these areas. One possibility is that strong mutual interactions within L4 could be important for spatial and pattern discrimination performance. The reciprocal connectivity between spiny stellate neurons (that are known to favor processing of local information[5]) and NMCs in S1 L4 might be related to the excellent performance on texture discrimination task with whiskers (25-μm-spaced particle texture discrimination[67,68]), which is better than that of humans with digits (75 μm[69]). In contrast, one-directional MC/PYR connectivity occurring in the apical dendrites in V1 likely serves a regulatory function by modulating feedback information coming from other layers and areas, and might be related to mice having a ~50 times lower high spatial frequency cutoff in visual contrast sensitivity compared to humans[70]. Interestingly, the principal excitatory cells in V1 L4 of cats[71] and monkeys[72], whose vision has higher spatial resolution vision, are also spiny stellate, whereas principal excitatory cells in S1 L4 of monkeys, who have worse tactile sense than rodents (250-μm-spaced particle texture discrimination[69]), are star pyramidal[73]. The L4 of the primary auditory cortex in cats is also dominated by pyramidal cells[74] and further work is needed to compare it to auditory cortices in other species. We hypothesize that whenever an L4 features spiny stellate cells, they might be reciprocally connected to local NMCs.

In this work we have focused on morphological cell types. At the same time, there is a growing understanding that cell-type definitions should take into account multimodal information, such as morphology, electrophysiology, and transcriptomics, as opposed to being based on a single modality[75]. In our V1 L4 dataset, we identified seven morphological types of interneurons but only four electrophysiological types (Fig. 2): four PV+ types could not be distinguished on the basis of their firing as they were all fast-spiking. This is in qualitative agreement with the findings of Gowens et al.[76], who, in a parallel work based on unsupervised clustering, identified twice as many morphological types (m-types) as electrophysiological ones (e-types). We only obtained the transcriptomic information for SOM+ neurons, but found that MCs in V1 L4 belonged to two different transcriptomic types (t-types) with slightly different electrophysiology (Fig. 4). On the other hand, we found that one of the SOM+ t-types (*Sst Hpse Cbln4*) exhibited markedly different morphology and electrophysiology depending on the cortical area. To the best of our knowledge, this is the first such example in the literature. It

suggests that the t-types cannot be taken as the ground truth for cell types and highlights the need for multimodal cell-type definitions.

In conclusion, we confirmed the difference in morphology of L4 principal cells and revealed another striking cellular difference between V1 and S1 of adult mice: morphology of L4 SOM+ interneurons. In each area, the morphology of SOM+ interneurons matched that of the excitatory neurons, suggesting that one of them might adapt to another. Furthermore, we found differences in the connections from excitatory neurons to SOM+ interneurons, suggesting a different functional role of SOM+ interneurons in different cortical areas. These results support the view that cell-type-specific circuit motifs, such as the Martinotti/pyramidal and non-Martinotti/stellate pairs, are used as building blocks to assemble the neocortex. In addition, our data suggest that the same transcriptomic cell type can exhibit different morphological and electrophysiological phenotypes depending on the cortical area, highlighting the need of multimodal profiling of cell types in the neocortex.

## Methods

**Animals**. Experiments on adult male and female mice (median age 72, interquartile range 63–88, full range 50–330 days, Supplementary Fig. 1) were performed using wild-type ($n = 24$), Viaat-Cre/Ai9 ($n = 47$), Scnn1a-Cre/Ai9 ($n = 5$ for V1 and $n = 5$ for S1), SOM-Cre/Ai9 (somatostatin, $n = 14$ for V1 and $n = 19$ for S1), VIP-Cre/Ai9 (vasoactive intestinal polypeptide, $n = 8$), and PV-Cre/Ai9 mice (parvalbumin, $n = 31$). Crossing Viaat-Cre mice (Viaat encodes a transporter required for loading GABA and glycine) with Ai9 reporter mice globally labels GABAergic interneurons with the fluorescence marker tdTomato[38]. SOM-Cre/Ai9 mice, VIP-Cre/Ai9 mice, and PV-Cre/Ai9 mice have SOM+ interneurons, PV+ interneurons, and VIP+ interneurons labeled with the fluorescent marker tdTomato respectively. Scnn1a-Cre/Ai9 mice preferentially have excitatory neurons in L4 labeled with tdTomato. Additional younger Scnn1a-Cre/Ai9 mice (P15-20, $n = 5$; P30-40, $n = 5$) were used to study connectivity between excitatory neurons at the different ages. Additional Scnn1a-Cre/Ai9 mice ($n = 8$) were used for measuring within-barrel connectivity between excitatory neurons in S1. Additional SOM-Cre/Ai9 mice ($n = 6$) were used for Patch-seq experiments. Animal preparation procedures for animals maintenance and surgeries were performed according to protocols approved by the Institutional Animal Care and Use Committee (IACUC) of Baylor College of Medicine.

The Viaat-Cre line was generously provided by Dr. Huda Zoghbi's laboratory. The other Cre lines were purchased from Jackson Laboratory:

- SOM-Cre: http://www.jax.org/strain/013044;
- VIP-Cre: http://www.jax.org/strain/010908;
- PV-Cre: http://www.jax.org/strain/008069;
- Scnn1a-Cre: https://www.jax.org/strain/013044;
- Ai9 reporter: http://www.jax.org/strain/007909.

**Slice preparation**. Slice preparation followed methods previously described by Jiang et al.[29]. Briefly, animals were deeply anesthetized using 3% isoflurane. After decapitation, the brain was removed and placed into cold (0–4 °C) oxygenated NMDG solution containing 93 mM NMDG, 93 mM HCl, 2.5 mM KCl, 1.2 mM NaH$_2$PO$_4$, 30 mM NaHCO$_3$, 20 mM HEPES, 25 mM glucose, 5 mM sodium ascorbate, 2 mM Thiourea, 3 mM sodium pyruvate, 10 mM MgSO$_4$ and 0.5 mM CaCl$_2$, pH 7.35 (all from Sigma–Aldrich). 300-μm-thick parasagittal slices were cut and special care was taken to select only slices that had a cutting plane parallel to the apical dendrites to ensure preservation of both axonal and dendritic arborization structures. The slices were incubated at 34.0 ± 0.5 °C in oxygenated NMDG solution for 10–15 min before being transferred to the artificial cerebrospinal fluid solution (ACSF) containing 125 mM NaCl, 2.5 mM KCl, 1.25 mM NaH$_2$PO$_4$, 25 mM NaHCO$_3$, 1 mM MgCl$_2$, 25 mM glucose and 2 mM CaCl$_2$, pH 7.4 (all from Sigma–Aldrich) for about 1 h. During recordings, slices were continuously perfused with oxygenated physiological solution throughout the recording session.

**Electrophysiological recordings**. Recordings were performed using glass pipettes (5–8 MΩ) filled with intracellular solution containing 120 mM potassium gluconate, 10 mM HEPES, 4 mM KCl, 4 mM MgATP, 0.3 mM Na3GTP, 10 mM sodium phosphocreatine and 0.5% biocytin, pH 7.25 (all from Sigma–Aldrich). We used two Quadro EPC 10 amplifiers (HEKA Elektronik, Lambrecht, Germany) that allowed us to perform simultaneous recordings up to eight cells. The PatchMaster software (HEKA Elektronik) and custom-written Matlab-based programs were used to operate the Quadro EPC 10 amplifiers and perform online and offline analysis of the data. In order to extract information about passive membrane properties and uncover the firing patterns, membrane potential of each neuron in

response to 600-ms-long current pulse injections were recorded (starting from −100/−200 pA with 20 pA increment step).

To identify synaptic connections, current pulses were injected into the presynaptic neurons (2 nA for 2 ms at 0.01–0.1 Hz) to evoke AP while the membrane potential of other simultaneously recorded neurons were monitored to detect unitary inhibitory or excitatory postsynaptic potentials (uI(E)PSPs). The uIPSPs were measured while the membrane potentials of the putative postsynaptic cells were held at −60 ± 3 mV, whereas uEPSPs were measured while membrane potentials of the putative postsynaptic cells were held at −70 ± 3 mV. Paired-pulse ratio (PPR) was calculated as the ratio between the mean amplitude of the second and the first uI(E)PSP obtained by injecting the presynaptic neuron with two consecutive stimuli of 2 nA with 100 ms interval. We recorded 10–30 individual traces and averaged the obtained uI(E)PSP amplitudes.

Neurons were assigned to L4 based on the neocortical layer boundaries and the small somata that characterize this layer, which were clearly visible in the micrograph under the bright-field microscope. The layer identity of each neuron was also confirmed by the visualization of their position after the staining.

Because the synaptic connectivity strongly depends on the inter-soma distance[29], we took special care to record from groups of neurons with inter-soma distances less than 150 μm. To make sure that the identified connections were monosynaptic, we recovered the morphology of the presynaptic neurons and made sure that the morphology and electrophysiology of the presynaptic neuron for each connection (i.e., pyramidal neurons vs. interneurons) matched the nature of connections (i.e., EPSP vs. IPSP). Indeed, the recovered morphology (i.e., pyramidal neurons vs. interneurons) and EPSP vs. IPSP always matched. Typical recording depth was 15–60 μm, similar to previous studies[29,65,77].

Importantly, neuronal structures can be severed (a limitation of all slice electrophysiology experiments) due to the slicing procedure, introducing a potential underestimation of neuronal morphology and connectivity. However, this did not seem to strongly affect the studies of local circuits in the past[29,78].

**Staining and morphology recovery**. Once the patch-clamp recording was terminated, the slices were immediately fixed by immersion in freshly-prepared 2.5% glutaraldehyde (from Electron Microscopy Science Cat.no. 16220), 4% paraformaldehyde (from Sigma–Aldrich Cat.no. P6148) in 0.1 M phosphate buffer at 4 °C for at least 72 h. The slices were subsequently processed with the avidin-biotin-peroxidase method in order to reveal the morphology of the neurons. To increase the success rate in recovering the morphology of GABAergic interneurons, especially the detail of their fine axonal arbors, we made additional modifications as described previously[29,30]. The morphologically recovered cells were examined and reconstructed using a ×100 oil-immersion objective lens and a camera lucida system (MicroBrightField, Vermont). Tissue shrinkage due to the fixation and staining procedures was about 10–20%, consistent with previous studies[29,51]. The shrinkage was not compensated for the morphology visualization and analysis.

To identify the barrels in S1 and relate their locations to tdTomato signal in Scnn1a-Cre mice, we performed cytochrome C staining brain slices from Scnn1a-Cre mice following protocols described in the literature[4,79]. To find the barrel locations using cytochrome C and tdTomato signals (Supplementary Fig. 12a), we averaged the pixel intensities as a function of horizontal position within the L4. The resulting intensity trace was normalized to lie between 0 and 1 and high-pass filtered to compensate for the uneven brightness of the images. To do the high-pass filter, we used a Fourier function of the form:

$$y = a_0 + a_1 \cos(wx) + b_1 \sin(wx) + a_2 \cos(2wx) + b_2 \sin(2wx) \qquad (1)$$

that was fitted to each trace ($w$ was fitted along the $a_i$ and $b_i$ coefficients) and then subtracted from it. The signal from the cytochrome C was inverted to match the directionality of the tdTomato signal. Barrel center locations were estimated as the positions of the peaks after smoothing with a σ = 250 μm Gaussian filter.

**Patch-seq procedure and sequencing**. To simultaneously obtain electrophysiology and transcriptome data from the same neurons, we applied our recently developed Patch-seq protocol[30] with minor modifications. Briefly, after careful cleaning the equipment and work surfaces with RNAse Zap[30], we prepared 300-μm-thick brain sections as described above. Recording pipettes of ~5 MΩ resistance were filled with 0.1–0.3 μL of RNase-free intracellular solution containing: 101 mM potassium gluconate, 4 mM KCl, 10 mM HEPES, 0.2 mM EGTA, 4 mM MgATP, 0.3 mM Na₃GTP, 5 mM sodium phosphocreatine (all from Sigma–Aldrich), and 1 U/μl recombinant RNase inhibitor (Takara Cat.no. 2313 A), pH ~ 7.25. This solution was slightly modified compared to the one described in Cadwell et al.[30] in order to obtain the osmolarity 300–320 mOSM without further water dilutions. During the recordings and sample collection, great care was taken to maintain an RNase-free environment by cleaning any items that become contaminated (such as the electrode wire if it comes into contact with ACSF) and changing gloves frequently. Electrophysiological recordings were performed as described above for multi-patch experiments. At the end of the recording, the cell contents were aspirated into the patch pipette by applying a gentle negative pressure (0.7–1.5 psi) for 2–10 min until the size of the cell body was visibly reduced. Soma structure and electrophysiological properties were constantly monitored during aspiration to ensure that the cell was healthy, and breaks would be taken if the cell appeared unstable or unhealthy. Special attention was taken to ensure that the seal between the pipette and the cell membrane was

intact during the entire procedure in order to avoid possible contamination from the extracellular environment. The contents of the pipette were immediately ejected into a 0.2 mL PCR tube containing 4 μL lysis buffer as described in Cadwell et al.[30]. The RNA was converted into cDNA using a Smart-seq2-based protocol[80] following the procedures described in detail in Cadwell et al.[30]. The size distribution and concentration of the cDNA libraries were analyzed using an Agilent Bioanalyzer 2100. cDNA samples containing less than 1 ng total cDNA (in the 15 μL of the final volume), or with an average size less than 1500 bp were not sequenced.

To construct the final sequencing libraries, 0.2 ng of purified cDNA from each sample was tagmented using the Illumina Nextera XT Library Preparation with one fifth of the volumes stated in the manufacturer's recommendation. The DNA was sequenced from single end (75 bp) with standard Illumina Nextera i5 and i7 index primers (8 bp each) using an Illumina NextSeq500 instrument. Investigators were blinded to cell type of Patch-seq samples during library construction and sequencing.

Reads were aligned to the mouse genome (mm10 assembly) using STAR (v2.4.2a) with default settings. We only used read counts (and not RPKM values, number of reads per kilobase of transcript per million total reads) for all data analysis presented here, but for completeness we mention that RPKM values were computed using rpkmforgenes[81] and NCBI RefSeq gene and transcript models (downloaded on the 24th of June 2014).

**Data analysis of the morphological reconstructions**. Reconstructed morphologies of $n = 92$ cells were converted into SWC format and further analyzed using custom Python code. Each cell was soma-centered and all neurites were smoothed in the slice depth dimension (Y) using a Savitzky-Golay filter of order 3 and window length 21, after resampling points to have 1 μm spacing. For further analysis we computed two different feature representations of each cell: the XZ density map (where XZ is the plane orthogonal to the slice depth; Z corresponds to cortical depth) and a set of morphometric statistics.

To compute the XZ density map, we sampled equidistant points with 100 nm spacing along each neurite and normalized the resulting point cloud such that the smallest coordinate across all points of all cells was 0 and the largest was 1[82]. The normalized point cloud was projected onto the xz-plane and binned into $100 \times 100$ bins spanning $[-0.1, 1.1]$. We smoothed the resulting density map by convolving it with an $11 \times 11$ bin Gaussian kernel with standard deviation $\sigma = 2$. For the purposes of downstream analysis, we treated this as a set of 10,000 features.

For each cell we computed a set of 16 summary statistics[45]: number of branch points, cell width, cell depth, cell height, number of tips, number of stems, total neurite length, maximal neurite length, maximum branch order, maximal segment length, average tortuosity, maximal tortuosity, average branch angle, maximal branch angle, average path angle, and maximal path angle.

We followed the classification approach that we recently benchmarked in Laturnus et al.[45]. As predictors for pairwise classification we used morphometric statistics and density maps. Due to the very high dimensionality of the density maps, we reduced them to 10 principal components (for cross-validation, PCA was computed on each outer-loop training set separately, and the same transformation was applied to the corresponding outer-loop test set). This makes the final feature dimensionality equal to 36.

For classification, we used logistic regression regularized with elastic net. Regularization parameter alpha was fixed to 0.5, which is giving equal weights to the lasso and ridge penalties. We used nested cross-validation to choose the optimal value of the regularization parameter lambda and to obtain an unbiased estimate of the performance. The inner loop was performed using the civisanalytics Python wrapper around the glmnet library[83] that does K-fold cross-validation internally. We used five folds for the inner loop. We kept the default setting which uses the maximal value of lambda with cross-validated loss within one standard error of the lowest loss (lambda_best) to make the test-set predictions:

```
LogitNet (alpha = 0.5, n_splits = 5, random_state = 42)
```

Note that the default behavior of glmnet is to standardize all predictors. The outer loop was 10 times repeated stratified five-fold cross-validation, as implemented in scikit-learn by

```
RepeatedStratifiedKFold (n_splits = 5, n_repeats = 10,
random_state = 43)
```

Outer-loop performance was assessed via test-set accuracy.

For the t-SNE visualization, we reduced density maps and morphometric statistics of the $n = 92$ cells to 10 principal components each. We scaled each set of 10 PCs by the standard deviation of the respective PC1, to make three sets be roughly on the same scale. Then we stacked them together to obtain a 20-dimensional representation of each cell. Exact (non-approximate) t-SNE was run with perplexity 15, random initialization with seed 42, and early exaggeration 4, using scikit-learn implementation:

```
TSNE (perplexity = 15, method = 'exact', random_state =
42, early_exaggeration = 4)
```

**Automatic extraction of electrophysiological features**. Thirteen electrophysiological features were automatically extracted using Python scripts from the

Allen Software Development Kit with some modifications to account for our experimental paradigms. An illustration of the feature extraction procedure for one exemplary neuron is shown in Supplementary Fig. 4. Here we briefly specify how each feature was extracted.

The resting membrane potential and the input resistance were computed differently for the standard patch-clamp/morphology recordings and for the Patch-seq recordings, because of the differences in the stimulation protocol between these two sets of experiments. In the Patch-seq experiments, the holding current before current step stimulation was fixed at 0 pA for all cells. Consequently, we computed the resting membrane potential as the median membrane voltage before stimulation onset. For each hyperpolarizing current injection, input resistance was calculated as the ratio of the steady state voltage deflection to the corresponding injected current value (we took the average voltage of the last 100 ms before stimulus offset as the steady state value). We took the median of these values as the final input resistance value. In contrast, in the standard patch-clamp experiments, the holding current before current step stimulation was not always fixed at 0 pA. For that reason we used linear regression (for robustness, random sample consensus regression, as implemented in `scikit-learn`) of the steady state membrane voltage onto the injected current value to compute the input resistance (regression slope) and the resting membrane potential (regression intercept) (Supplementary Fig. 4d). For this we used five highest hyperpolarizing currents (if there were fewer than five, we used those available).

To estimate the rheobase (the minimum current needed to elicit any spikes), we used robust regression of the spiking frequency onto the injected current using the five lowest depolarizing currents with non-zero spike count (if there were fewer than five, we used those available) (Supplementary Fig. 4d). The point where the regression line crosses the x-axis gives the rheobase estimate. We restricted the rheobase estimate to be between the highest injected current eliciting no spikes and the lowest injected current eliciting at least one spike. In the rare cases when the regression line crossed the x-axis outside of this interval, the nearest edge of the interval was taken instead as the rheobase estimate.

The action potential (AP) threshold, AP amplitude, AP width, afterhyperpolarization (AHP), afterdepolarization (ADP), and the first AP latency were computed as illustrated in Supplementary Fig. 4c, using the very first AP fired by the neuron. AP width was computed at the half AP height.

The adaptation index (AI) is defined as the ratio of the second interspike interval to the first one (Supplementary Fig. 4b). We took the median over the five lowest depolarizing currents that elicited at least three spikes (if fewer than five were available, we used all of them).

The maximum number of APs simply refers to the maximum number of APs emitted in the 600 ms stimulation window overall stimulation currents (Supplementary Fig. 4a). The membrane time constant (tau) was computed as the time constant of the exponential fit to the first phase of hyperpolarization (the median overall hyperpolarizing traces). Finally, the sag ratio is defined as the ratio of the maximum membrane voltage deflection to the steady state membrane voltage during the first (the lowest) hyperpolarizing current injection.

**Data analysis of the electrophysiological features**. For the t-SNE visualization (Fig. 2b), we log-transformed the AI values because this feature had a strongly right-skewed distribution (Supplementary Fig. 5). We also excluded ADP and latency; ADP because it was equal to zero for most neurons and rare cells with non-zero values appeared as isolated subpopulations in the t-SNE representation, and latency because it had high outliers among the FS types, also yielding isolated subpopulations. The remaining 11 features were z-scored and exact (non-approximate) t-SNE was run with perplexity 15 and random initialization with seed 42 using `scikit-learn` implementation: `TSNE (perplexity = 15, method = 'exact', random_state = 42)`

For pairwise classification, we used exactly the same procedure as described above for pairwise classification using the reconstructed morphologies (nested cross-validation with logistic regression regularized with elastic net). All 13 features were used, with log-transformed AI and log-transformed latency (as shown in Supplementary Fig. 5).

**Data analysis of the RNA-seq data**. The total number of sequenced cells was $n = 118$. Four cells were excluded because the sum of counts across all genes was below 1500 (Supplementary Fig. 7a). The remaining $n = 114$ cells were mapped to the full set of 133 transcriptomic types identified in Tasic et al.[25]; see below for the details. One cell was excluded because it mapped to one of the excitatory types, and three cells were excluded because they mapped to *Pvalb Reln Itm2a* type (and were fast-spiking). All the remaining $n = 110$ cells mapped to the *Sst* types. Among those, eight cells did not have good electrophysiological recordings (the recordings were either lost or were of bad quality) and were excluded from all downstream analyses that required electrophysiological data (leaving $n = 102$ cells).

The mapping to the reference types was done as follows. Using the count matrix of Tasic et al. ($n = 23,822$, $d = 45,768$), we selected 3000 most variable genes (see below). We then log-transformed all counts with $\log_2(x + 1)$ transformation and averaged the log-transformed counts across all cells in each of the 133 clusters, to obtain reference transcriptomic profiles of each cluster ($133 \times 3000$ matrix). Out of these 3000 genes, 2686 were present in the `mm10` reference genome that we used to align reads in our data (see above). We applied the same $\log_2(x + 1)$ transformation

to the read counts of our cells, and for each cell computed Pearson correlation across the 2686 genes with all 133 Tasic et al. clusters. Each cell was assigned to the cluster to which it had the highest correlation.

To select the most variable genes, we found genes that had, at the same time, high non-zero expression and high probability of near-zero expression[84]. Our procedure is described in more detail elsewhere[85]. Specifically, we excluded all genes that had counts of at least 32 in fewer than 10 cells. For each remaining gene we computed the mean $\log_2$ count across all counts that were larger than 32 (non-zero expression, $\mu$) and the fraction of counts that were smaller than 32 (probability of near-zero expression, $\tau$). Across genes, there was a clear inverse relationship between $\mu$ and $\tau$, that roughly followed exponential law $\tau \approx \exp(-1.5 \cdot \mu + a)$ for some horizontal offset $a$. Using a binary search, we found a value $b$ of this offset that yielded 3000 genes with $\tau > \exp(-1.5 \cdot \mu + b) + 0.02$. These 3000 genes were selected.

The t-SNE visualization of the whole Tasic et al.[25] dataset shown in Supplementary Fig. 7c was taken from our previous work[85]. It was computed there using scaled PCA initialization, perplexity combination of 30 and 238 (1% of the sample size), and learning rate 23,822/12, following preprocessing steps of sequencing depth normalization (by converting counts to counts per million), feature selection (3000 most variable genes), $\log_2(x + 1)$ transformation, and reducing the dimensionality to 50 using PCA.

To make t-SNE visualization of the somatostatin part of the Tasic et al. dataset (Fig. 4b), we selected all cells from all *Sst* types apart from the very distinct *Sst Chodl* (20 types, 2701 cells). Using these cells, we selected 500 most variable genes using the same procedure as described above. We used the same preprocessing steps as above, perplexity 50, and scaled PCA initialization[85].

For each of the $n = 110$ Patch-seq cells, we computed its Pearson correlation with each of the 2701 reference cells across the 500 genes, most variable in the somatostatin part of the Tasic et al. dataset (only 472 genes present in our data were used). Then we found 10 reference cells with the highest correlations (10 nearest neighbours of the Patch-seq cell) and positioned our cell at the coordinate-wise median t-SNE location of those 10 reference cells[85].

The mapping of the $n = 110$ Patch-seq cells to the 20 somatostatin types (Fig. 4c) was done exactly as the mapping to the full set of 133 clusters described above, but this time only using 500 genes, most variable in the somatostatin part of the Tasic et al.[25] dataset (only 472 genes present in our data were used).

We used our implementation of sparse reduced-rank regression (RRR) described in detail elsewhere[48]. For the analysis shown in Fig. 4e, we selected 1000 most variable genes as described above, using $n = 102$ Patch-seq cells with high-quality electrophysiological recordings. The gene counts were converted to counts per million and $\log_2(x + 1)$-transformed. The columns of the resulting $102 \times 1000$ expression matrix were standardized. All 13 electrophysiological features (AI and latency log-transformed) were standardized as well. The rank of RRR was fixed at 2. We used 10-fold cross-validation to select the values of alpha and lambda regularization parameters that would maximize the predicted R-squared. This yielded alpha $= 0.5$ and lambda $= 1$ (with relaxed elastic net[48]). Fig. 4e shows scatter plots of the two standardized RRR components in the transcriptomic and in the electrophysiological spaces. Features and genes are depicted as lines showing correlations of a feature/gene with each of the two components. In the electrophysiological space, all features are shown. In the transcriptomic space, only genes selected by the model are shown. The values of R-squared and correlations between the components from electrophysiological and transcriptomic spaces reported in the caption of Fig. 4e are cross-validation estimates.

## Data availability

Sequencing data are available under accession number GSE134378. Apart from the raw reads, this link contains a table of read counts and a table of RPKM values. Raw electrophysiological recordings (in .mat format) and morphological reconstructions (in .asc and .swc formats) are deposited to Zenodo at https://doi.org/10.5281/zenodo.3336165. Morphological reconstructions will also be made available at http://neuromorpho.org in archive "Tolias".

## Code availability

The analysis code in Python is available at http://github.com/berenslab/layer4. This includes data analysis of electrophysiological recordings, data analysis of the morphological reconstructions, and data analysis of the transcriptomic data. This repository also includes a table of the extracted electrophysiological features for the morphological and for Patch-seq data sets.

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

## Acknowledgements

We thank Alexander Naka for comments and suggestions. Supported by the Intelligence Advanced Research Projects Activity (IARPA) via Department of Interior/Interior Business Center (DoI/IBC) contract number D16PC00003. The U.S. Government is authorized to reproduce and distribute reprints for Governmental purposes notwithstanding any copyright annotation thereon. The views and conclusions contained herein are those of the authors and should not be interpreted as necessarily representing the official policies or endorsements, either expressed or implied, of IARPA, DoI/IBC, or the U.S. Government. This work was also supported by the National Institute of Mental Health under award numbers U19MH114830, R01 MH109556, T32EY00252037, and the NSF NeuroNex program through grant NSF-1707359. The content is solely the responsibility of the authors and does not necessarily represent the official views of the National Institutes of Health. This work was also supported by the German Research Foundation (BE5601/4-1, SFB1233 - 276693517), the German Excellence Strategy (EXC 2064 - 390727645), and the Federal Ministry of Education and Research (FKZ 01GQ1601).

## Author contributions

F.S. and X.J. performed electrophysiological recordings and manual neuronal reconstructions. D.K. supervised data analysis. J.C. created full-length cDNA libraries and aided in morphological recovery. L.H. prepared the full-length cDNA libraries for sequencing and performed initial bioinformatics analysis under the supervision of R.S. Z.T. and S.P. sustained animals' colonies and provided experimental support. S.L. did the morphological data analysis. Y.B. did the electrophysiological data analysis. D.K. did the transcriptomic data analysis. F.S., D.K., S.L., Y.B. and E.F. analyzed the data and produced the figures. F.S., D.K., S.S., C.R.C., P.B., E.F., X.J. and A.S.T. wrote the manuscript. A.S.T., X.J., S.S.P. and P.B. discussed and oversaw analysis and results. All authors revised the manuscript.

## Additional information

**Competing interests:** The authors declare no competing interests.

