## [Transparent Peer Review File · Nature Communications]

Reviewers' comments:

Reviewer #2 (Remarks to the Author):

Scala et al sought to investigate neuronal diversity in layer 4 of two primary functional areas of the cortex, the visual cortex and the somatosensory cortex. Overall, the study is carefully conducted and involves a substantial amount of novel data, with a very specific focus on one layer of the mouse cerebral cortex in two cortical areas. I think there a real value in this study for the resource it provides. However, I would urge the authors to find more accessible ways to communicate their findings. Currently, the results descriptions are very detailed, and very hard to follow for non-experts.

Initially, the authors used normal, wild type mice to randomly sample layer 4 neurons and found vast majority to be pyramidal neurons, with a minor fraction of stellate neurons. The authors then used several of the widely used cre mouse lines to visualize and quantify cellular morphologies, and described quantitative estimates of each morphological cell type labelled with this approach.

Subsequently, the authors use regularized logistic regression on morphological and electrophysiological features to evaluate the utility of these features in discriminating between cell types. Overall, the study is very descriptive, and somewhat difficult to follow. Many of the findings, including composition differences in spiny stellate cells across areas have been noted before, and even some of the connectivity probabilities are largely known.

One novel discussion is that of interneuron diversity, where among Martinotti cells, transcriptomically similar subtypes across areas may adopt different morphologies, and have slightly different electrophysiological properties. This is consistent with the idea that interneurons exhibit some degree of plasticity and may adopt more diverse states depending on activity and niche environment. What is new is the idea that some of the molecular features may not be entirely determined by these factors, and I think this is worth highlighting. This finding contrasts that of Basket cells which appear to largely preserve their connectivity preferences across areas.

The authors summarize the intellectual contribution by suggesting that an organizing principle in the cortex may be that of preservation of cell pairs, including for example non-Martinotti/stellate versus Martinotti/pyramidal pairs. One issue I have is that this model is based largely on a correlation, but they don't establish causation.

Reviewer #4 (Remarks to the Author):

The paper by Tolias and colleagues takes a comprehensive looks at the composition of L4 cells in visual versus somatosensory cortex and notes that there is a matching in the presence of stellate versus pyramidal neurons compared to non-Martinotti versus Martinotti neurons, respectively.

Incidentally, it is my understanding that in higher mammals that stellate cells dominate in areas of cortex beyond S1 and are plentiful in V1. If I am correct, the authors should at very least comment on these species differences and emphasize that the present work reflects what is seen in rodents but not carnivores or primates.

The classification used by the Tolias group is somewhat confusing and at odds with the general literature. First they split FS cells into four categories but describe BC separately from what they describe as SC, DBC and HEC cells. First separating BC from the rest is confusing and the presence of the remaining three classes of FS cells morphologically is at odds with the bulk of the literature. Interestingly, they make no similar distinction between the Martinotti neurons they examine, which clearly include both T-shaped and fanning subtypes. It would be interesting to know if these are divided into LTS and rebound bursting subtypes but their rather cursory analysis of their physiology stops short of investigating this question. This seems at odds with their later focus on the differences between S1 and V1 SST interneurons.

The paper next turns to the matching of NMC with stellate cells in S1 versus MC with pyramidal cells in V1. This is the heart of what makes this paper interesting. In particular the realization that there exists a mismatch between MC at NMC cells at a gene expression level but a mismatch in morphology and physiological properties. They attempt to show that this mismatch can be understood as these cells falling along a gradient of differences in gene expression. This perhaps is true but in my mind this is a missed opportunity to embrace the seeming reality that common gene expression can still manifest in different phenotypes in terms of morphology and physiology, which is the real take home message.

The paper is a wonderful collection of data accompanied by a very nice quantitative analysis that extends the use of unsupervised clustering to morphology in a manner that should be adopted widely. That said the desire to make this paper both an exhaustive effort in revising cell classification methods buries the key finding of the paper which is that there is a clear matching between excitatory and inhibitory types, with excitatory type predicting (and perhaps determining) the morphology, connectivity and physiology of their partners BUT not their gene expression. This emphasizes the post-transcriptional influences that have been underappreciated and should not in my opinion be undercut by the more global classification message of the paper. I would encourage the authors therefore to streamline their paper and publish details of the cataloging of cell types and the methods to analyze them elsewhere and focus here on the MC/pyramidal vs NMC/stellate, which I find of paramount biological importance. Either way this is an excellent paper whose publication I fully support.

Reviewer #5 (Remarks to the Author):

Review of Scala et al. "Neocortical layer 4 in adult mouse differs in major cell types and circuit organization between primary sensory areas"

The work of Scala et al. uses a cutting-edge array of techniques (multiple patch clamp recordings, morphological reconstruction, single cell RNA-seq and computational methods) to comparatively study the connectivity of inhibitory but also excitatory neurons in layer IV of V1 and S1BF. On the one hand, it is a follow-up of the impressive 2015 Science publication (Jiang et al.) and fills the gap of missing granular layer circuitry and on the other hand it presents a highly interesting area-specific connectivity, which challenges the idea that neocortex is wired in a stereotypic, region-independent manner. An additional strength of this paper is that it presents original data for the previous claim (in Jiang et al) that the strong difference in terms (of close to zero) connection probability for excitatory neurons (in contrast to previous literature), might be an age-related effect. Despite all these achievements, I have a number of comments and suggestions that aim to make the paper stronger and thus adequate for publication.

There are two major issues I have with the manuscript.

1.) Number of inhibitory cell types and their names. The interneuron field is plagued by a cacophony of names that are either misleading or "invent" the same cell type multiple times.

Although I realized that naming is a complicated business, I would like to make some remarks here. For the excitatory cells, the authors have chosen to restrict themselves to distinguish stellate from pyramidal cells. The more precise term for the first is "spiny stellate cell" (to disambiguate them from the smooth stellate cells) and the latter (in layers IV and VI) often appear as the variation of a star pyramidal cell (e.g. the excitatory cell in Fig. 5a), which might have a different connectivity profile (Schubert et al., 2003; Staiger et al, 2004; Stricker and Cowan, 2004), at least in the barrel cortex. For the inhibitory cells, I am completely fine with BC, MC, BPC and NGC. However, the remaining ones (all being FS!) are potentially BC variants that are now disguised with figurative names, the figure of which can be very arbitrary. Ever since Cajal, it is unclear what a DBC is. 2 bunchs of dendrites, 2 bunchs of axons or 1 bunch of dendrite and 1 of axon? What is a shrub cell? It looks like a focally-projecting BC, similar to what has been described in Koelbl et al., 2015 or Munoz et al., 2014. The HEC cells in LVA look like layer-adapted BC, if they really also exist in LIV (which I would like to see by an image clearly showing that), they might be "FS transcolumar inhibitors", adding to the observations of Emmenegger et al. 2018. Please reconsider your terminology and also discuss this issue.

2.) The paucity of excitatory connections. The original idea that pyramidal cells do the computations and interneurons are slave to them, is certainly not true. However, simply by their numbers and the fact that without connections there is no computation, excitatory neurons should be much more strongly connected than what has been found in Scala et al. (and Jiang et al.). The time course study is a great eye-opener, especially since it also explains discrepancy to Seeman et al., 2018, but I would like to suggest to consolidate that issue by doing a sufficient number of control experiments with a cesium-based intracellular solution to reduce the space clamp problem of whole cell recordings from the soma.

3.) It would be helpful that have a comprehensive connectivity matrix in the main paper and not only a partial one in Fig. S10.

Other points:

1.) Abstract, l. 29: optionally sounds so arbitrary. Such specific changes should be "by design" and have specific purpose for the computation of sensory information, which seems to have different needs in V1 versus S1BF.

2.) L. 131: vasoactive intestinal polypeptide is the most frequently used term.

3.) Figure 5: electrophysiological traces are much too small.

4.) Discussion: It starts " like a bull at a gate". I would prefer a start that somehow tries to summarize and contrast our current understanding of L4 "function" in V1 versus S1. In the same direction, Discussion stops abruptly at l. 474. It would be great to hear a concise hypothesis of what the authors mean that could mean in terms of computing sensory information, which comes from distinct set of receptors in somatosensory but a continuous sheet of receptors in visual cortex.

5.) L. 508/509. Scnn1a is not selective for L IV, there is quite a strong infragranular component as well. It was recently quantified, unfortunately, I do not remember the reference.

6.) L. 618: what is tagmented?

We would like to thank the reviewers for their very helpful comments and criticisms of our manuscript. We believe we have addressed all the concerns, and in particular, have revised our cell type naming scheme. Please find below our point by point reply to the reviewers' comments.

Reviewer #2

Scala et al sought to investigate neuronal diversity in layer 4 of two primary functional areas of the cortex, the visual cortex and the somatosensory cortex. Overall, the study is carefully conducted and involves a substantial amount of novel data, with a very specific focus on one layer of the mouse cerebral cortex in two cortical areas. I think there a real value in this study for the resource it provides. However, I would urge the authors to find more accessible ways to communicate their findings. Currently, the results descriptions are very detailed, and very hard to follow for non-experts.

We thank the Reviewer for the positive comments. We have revised the manuscript and tried to make the presentation more accessible. After the previous round of review, we moved the detailed description of neural morphologies from the Results into the Supplementary Text. Now we have further condensed and streamlined the description of neural cell types. If the text requires further modification, we would be grateful for any specific suggestions about what parts of the presentation can be further improved.

Initially, the authors used normal, wild type mice to randomly sample layer 4 neurons and found vast majority to be pyramidal neurons, with a minor fraction of stellate neurons. The authors then used several of the widely used cre mouse lines to visualize and quantify cellular morphologies, and described quantitative estimates of each morphological cell type labelled with this approach.

Subsequently, the authors use regularized logistic regression on morphological and electrophysiological features to evaluate the utility of these features in discriminating between cell types. Overall, the study is very descriptive, and somewhat difficult to follow. Many of the findings, including composition differences in spiny stellate cells across areas have been noted before, and even some of the connectivity probabilities are largely known.

During the revision, we further edited the manuscript to make the presentation as accessible as possible. Regarding the known difference in prevalence of spiny stellate cells between S1 and V1, this was pointed out by Reviewer 1 in the previous round of review, and we have modified the text accordingly before the submission to Nature Communications. We have now additionally made sure that we are not presenting this as a novel finding.

One novel discussion is that of interneuron diversity, where among Martinotti cells, transcriptomically similar subtypes across areas may adopt different morphologies, and

have slightly different electrophysiological properties. This is consistent with the idea that interneurons exhibit some degree of plasticity and may adopt more diverse states depending on activity and niche environment. What is new is the idea that some of the molecular features may not be entirely determined by these factors, and I think this is worth highlighting. This finding contrasts that of Basket cells which appear to largely preserve their connectivity preferences across areas

We agree with the Reviewer that this is one of the central findings of our study. We have edited the Discussion to reinforce this idea (see subsections “Transcriptomic types of SOM⁺ interneurons in L4: V1 vs. S1”, “Towards multimodal cell type definition”, and “Summary”).

The authors summarize the intellectual contribution by suggesting that an organizing principle in the cortex may be that of preservation of cell pairs, including for example non-Martinotti/stellate versus Martinotti/pyramidal pairs. One issue I have is that this model is based largely on a correlation, but they don't establish causation.

We fully agree with the Reviewer that further studies are needed to understand the developmental origins of this pairing, but we believe this important question is beyond the scope of this paper since causal experiments will require a completely new study.

Reviewer #4

The paper by Tolias and colleagues takes a comprehensive look at the composition of L4 cells in visual versus somatosensory cortex and notes that there is a matching in the presence of stellate versus pyramidal neurons compared to non-Martinotti versus Martinotti neurons, respectively.

Incidentally, it is my understanding that in higher mammals the stellate cells dominate in areas of cortex beyond S1 and are plentiful in V1. If I am correct, the authors should at very least comment on these species differences and emphasize that the present work reflects what is seen in rodents but not carnivores or primates.

The Reviewer is correct. As we wrote in the beginning of the Discussion, “Excitatory neurons in V1 L4 of other species such as cats and monkeys are also known to be spiny stellate. It remains an open question, why pyramidal cells in rodent V1 L4 remain pyramidal, whereas L4 excitatory cells in rodent S1 and in V1 of other non-rodent species develop into spiny stellate cells.”

The classification used by the Tolias group is somewhat confusing and at odds with the general literature. First they split FS cells into four categories but describe BC separately from what they describe as SC, DBC and HEC cells. First separating BC from the rest is confusing and the presence of the remaining three classes of FS cells morphologically is

at odds with the bulk of the literature. Interestingly, they make no similar distinction between the Martinotti neurons they examine, which clearly include both T-shaped and fanning subtypes. It would be interesting to know if these are divided into LTS and rebound bursting subtypes but their rather cursory analysis of their physiology stops short of investigating this question. This seems at odds with their later focus on the differences between S1 and V1 SST interneurons.

The Reviewer raises an important issue that was also raised by Reviewer 5. Based on these critical comments we reconsidered our naming nomenclature, and have renamed BC/SC/DBC/HEC into “large basket cells”, “small basket cells”, “double-bouquet basket cells”, and “horizontally elongated basket cells”. Indeed, the literature suggests that PV⁺ FS cells are either basket or chandelier, and none of our cells had chandelier-like axonal structured. Based on that, we conclude that all our FS cells are indeed basket cells. At the same time, we believe that the morphology of these four types is distinct (Figure 1) and so want to separate them into four distinct morphological types (see also Jiang et al., 2015¹). The new naming convention can hopefully be a compromise because it highlights that all four types are under the umbrella of “basket cells”, and at the same time, the names correspond to the types identified in our previous Jiang et al., Science 2015¹, publication (in L2/3 and L5).

Regarding the Martinotti neurons, the Reviewer makes a good point. We now looked at the firing pattern of our MCs in V1 L4 and did not find a single cell with a rebound (we have inserted this statement into the Results). Also, we do not agree that the morphology of our MCs “clearly include both T-shaped and fanning subtypes”. The difference between T-shaped and fanning-out MCs previously identified in L5 of S1² seems much stronger than any differences we observe between MCs in L4 of V1. Altogether, we did not find evidence for splitting MC type into further subtypes.

The paper next turns to the matching of NMC with stellate cells in S1 versus MC with pyramidal cells in V1. This is the heart of what makes this paper interesting. In particular the realization that there exists a mismatch between MC at NMC cells at a gene expression level but a mismatch in morphology and physiological properties. They attempt to show that this mismatch can be understood as these cells falling along a gradient of differences in gene expression. This perhaps is true but in my mind this is a missed opportunity to embrace the seeming reality that common gene expression can still manifest in different phenotypes in terms of morphology and physiology, which is the real take home message.

We agree with the Reviewer. We have now edited the Introduction (last paragraph) and the Discussion to reinforce this idea (see subsections “Transcriptomic types of SOM⁺ interneurons in L4: V1 vs. S1”, “Towards a multimodal cell type definition”, and “Summary”). Also, we decided to remove from the Discussion the paragraph about gradient in expression because it was indeed somewhat distracting from the more important point.

The paper is a wonderful collection of data accompanied by a very nice quantitative analysis that extends the use of unsupervised clustering to morphology in a manner that should be adopted widely. That said the desire to make this paper both an exhaustive effort in revising cell classification methods buries the key finding of the paper which is that there is a clear matching between excitatory and inhibitory types, with excitatory type predicting (and perhaps determining) the morphology, connectivity and physiology of their partners BUT not their gene expression. This emphasizes the post-transcriptional influences that have been underappreciated and should not in my opinion be undercut by the more global classification message of the paper. I would encourage the authors therefore to streamline their paper and publish details of the cataloging of cell types and the methods to analyze them elsewhere and focus here on the MC/pyramidal vs NMC/stellate, which I find of paramount biological importance. Either way this is an excellent paper whose publication I fully support.

We thank the reviewer for the positive comments and the suggestion. We have considered the possibility of splitting the paper into two separate manuscripts, and have discussed it with the editor. In the end, we decided to keep it together. We thank the Reviewer for supporting the publication of our work.

Reviewer #5

The work of Scala et al. uses a multiple patch clamp recordings, morphological reconstruction, single cell RNA-seq and computational methods) to comparatively study the connectivity of inhibitory but also excitatory neurons in layer IV of V1 and S1BF. On the one hand, it is a follow-up of the impressive 2015 Science publication (Jiang et al.) and fills the gap of missing granular layer circuitry and on the other hand it presents a highly interesting area-specific connectivity, which challenges the idea that neocortex is wired in a stereotypic, region-independent manner. An additional strength of this paper is that it presents original data for the previous claim (in Jiang et al) that the strong difference in terms (of close to zero) connection probability for excitatory neurons (in contrast to previous literature), might be an age-related effect. Despite all these achievements, I have a number of comments and suggestions that aim to make the paper stronger and thus adequate for publication.

There are two major issues I have with the manuscript.

1.) Number of inhibitory cell types and their names. The interneuron field is plagued by a cacophony of names that are either misleading or "invent" the same cell type multiple times. Although I realized that naming is a complicated business, I would like to make some remarks here. For the excitatory cells, the authors have chosen to restrict themselves to distinguish stellate from pyramidal cells. The more precise term for the first is "spiny stellate cell" (to disambiguate them from the smooth stellate cells) and the latter (in layers IV and VI) often appear as the variation of a star pyramidal cell (e.g. the

excitatory cell in Fig. 5a), which might have a different connectivity profile (Schubert et al., 2003; Staiger et al, 2004; Stricker and Cowan, 2004), at least in the barrel cortex.

We thank the Reviewer for raising the issue of cell type nomenclature. We now made sure that we do not omit “spiny” from “spiny stellate” cells. Regarding the star pyramidal vs. pyramidal cells, indeed we decided not to make this distinction in this paper. We have now inserted a sentence into the Discussion to comment on that.

For the inhibitory cells, I am completely fine with BC, MC, BPC and NGC. However, the remaining ones (all being FS!) are potentially BC variants that are now disguised with figurative names, the figure of which can be very arbitrary. Ever since Cajal, it is unclear what a DBC is. 2 bunchs of dendrites, 2 bunchs of axons or 1 bunch of dendrite and 1 of axon? What is a shrub cell? It looks like a focally-projecting BC, similar to what has been described in Koelbl et al., 2015 or Munoz et al., 2014. The HEC cells in LVa look like layer-adapted BC, if they really also exist in LIV (which I would like to see by an image clearly showing that), they might be “FS transcolumar inhibitors”, adding to the observations of Emmenegger et al. 2018. Please reconsider your terminology and also discuss this issue.

This is in line with the criticism of Reviewer 4. Based on these critical comments we reconsidered our naming nomenclature, and have renamed BC/SC/DBC/HEC into “large basket cells”, “small basket cells”, “double-bouquet basket cells”, and “horizontally elongated basket cells”. Indeed, the literature suggests that PV⁺ FS cells are either basket or chandelier, and none of our cells had chandelier-like axonal structures. Based on that, all our FS cells are likely to be basket cells. At the same time, we believe that the morphology of these four types is very different (Figure 1) and so want to separate them into four distinct morphological types. The new naming convention can hopefully be a compromise because it highlights that all four types are under the umbrella of “basket cells”, and at the same time, the names correspond to the types identified in our previous paper ¹ reporting data from L2/3 and L5.

Regarding the horizontally elongated basket cells, we found this kind of cell morphology in L4. Here is one example, with the soma located at 360 microns below the pia: (a) acquired image at 4x magnification showing the horizontal cell together with other stained neurons; (b) the same image at 10x magnification that allow the visualization of horizontal projections; (c) a 2D manual reconstruction of the cell realized with NeuroLucida at 100x magnification.

2.) The paucity of excitatory connections. The original idea that pyramidal cells do the computations and interneurons are slave to them, is certainly not true. However, simply by their numbers and the fact that without connections there is no computation, excitatory neurons should be much more strongly connected than what has been found in Scala et al. (and Jiang et al.). The time course study is a great eye-opener, especially since it also explains discrepancy to Seeman et al., 2018, but I would like to suggest to consolidate that issue by doing a sufficient number of control experiments with a cesium-based intracellular solution to reduce the space clamp problem of whole cell recordings from the soma.

Based on this suggestion, we have reviewed the literature to determine the effectiveness of achieving iso-potential clamp in the entire cell via a single electrode voltage patching in the soma. A modeling study, in which intracellular Cesium was used, has shown that such an iso-potential clamp is extremely difficult to achieve in a leaky neuron (Fig. 5, ³) even after K⁺ channels are fully and uniformly blocked throughout the cell. Further, another modeling study has shown that increase in membrane resistance (one of the main effects of cesium) only reduced the steady-state voltage attenuation from dendrites to soma and had minimal influence on transient voltage attenuation (Fig. 1, ⁴, suggesting a minimal reduction of attenuation on brief postsynaptic dendritic voltage evoked by a presynaptic spike. Consistent with these modeling studies, Williams and Mitchell ^{4,5} found that intracellular cesium diffusion throughout the cell does not improve the measurement of synaptic currents at the voltage clamped soma.

Together, these studies suggest that even if we conduct the cesium experiment and determine the probability of connection between two types of neurons, we will not be able to conclusively determine their anatomical connectivity. On the other hand, the connectivity determined in a cesium experiment will not correspond to the functional connectivity under a normal synaptic function either. We think it is best to wait for an experimental procedure that would be able to create an iso-potential condition across the entire cell before undertaking these experiments and until then consider all estimated connectivities as functional (as opposed to anatomical).

Furthermore, we have reported that connectivity decreased with age from 13% (at P15) to 1%. The space-clamp problem could only be an explanation for that if there were reasons to believe

that space-clamp problem is more severe in adult animals and less severe in the young. However, the dendrite thickness/abundance remains approximately constant between P15 and P60, at least according to ⁶. Also, if the dendritic diameter is smaller in younger animals, then this should increase the signal attenuation. Based on these considerations, we do not think that space-clamp issue could provide a plausible explanation for an apparent connectivity decrease with age.

That said, we agree that the space clamp problems together with distortions introduced by cutting procedures in our study, as well as in most other studies of this kind, certainly can affect somatic measurements of currents from distally located synapses. The magnitude of this effect is difficult to estimate and we believe that only through a combination of other methods (such as *in-vivo* recordings and electron microscopy) we will be able to deeply investigate neuronal wiring diagrams in the brain. To this end we are collaborating with the AIBS and Princeton for a large scale electron microscopy study in mouse V1 which we hope will provide more accurate estimates of the anatomical connectivity between pyramidal types.

3.) It would be helpful that have a comprehensive connectivity matrix in the main paper and not only a partial one in Fig. S10.

We have now followed this suggestion and incorporated 3x3 connectivity matrix (earlier shown in Fig. S10) into Figure 5.

Other points:

1.) Abstract, l. 29: optionally sounds so arbitrary. Such specific changes should be “by design” and have specific purpose for the computation of sensory information, which seems to have different needs in V1 versus S1BF.

We have removed the word “optionally”.

2.) L. 131: vasoactive intestinal polypeptide is the most frequently used term.

Following the suggestion, we have modified the text.

3.) Figure 5: electrophysiological traces are much too small.

We have updated the Fig. 4 as suggested by the reviewer and made the electrophysiological traces larger.

4.) *Discussion: It starts "like a bull at a gate". I would prefer a start that somehow tries to summarize and contrast our current understanding of L4 "function" in V1 versus S1. In the same direction, Discussion stops abruptly at l. 474. It would be great to hear a concise hypothesis of what the authors mean that could mean in terms of computing sensory information, which comes from distinct set of receptors in somatosensory but a continuous sheet of receptors in visual cortex.*

We have inserted an introductory paragraph into the Discussion. We have also made a new subsection at the end of Discussion, called "Functional implications". There we speculate as to why L4 in rodent S1 may be different from L4 in rodent V1. Regarding the Reviewer's comment that it might be related to the difference between discrete and continuous receptors, we are not sure that this can explain the differences observed in other species (e.g. cats and monkeys have stellate neurons in V1, but monkeys have pyramidal neurons in S1). Rather, we hypothesize that the difference might be related to the "resolution" of the corresponding sensory modality.

5.) *L. 508/509. Scnn1a is not selective for L IV, there is quite a strong infragranular component as well. It was recently quantified, unfortunately, I do not remember the reference.*

We have reformulated this sentence to say that Scnn1a "preferentially labels excitatory neurons in L4". This does not affect our conclusions.

6.) *L. 618: what is tagmented?*

Tagmentation is the first step in library preparation for sequencing. To quote the Illumina website, "The first step in library preparation is the tagmentation reaction, which involves the transposon cleaving and tagging of the double-stranded DNA with a universal overhang." See also our earlier paper introducing the Patch-seq⁷.

References

1. Jiang, X. *et al.* Principles of connectivity among morphologically defined cell types in adult neocortex. *Science* **350**, aac9462–aac9462 (2015).
2. Nigro, M. J., Hashikawa-Yamasaki, Y. & Rudy, B. Diversity and Connectivity of Layer 5 Somatostatin-Expressing Interneurons in the Mouse Barrel Cortex. *The Journal of Neuroscience* **38**, 1622–1633 (2018).

3. Fleidervish, I. A. & Libman, L. How cesium dialysis affects the passive properties of pyramidal neurons: implications for voltage clamp studies of persistent sodium current. *New Journal of Physics* **10**, 035001 (2008).
4. Spruston, N., Jaffe, D. B., Williams, S. H. & Johnston, D. Voltage- and space-clamp errors associated with the measurement of electrotonically remote synaptic events. *Journal of Neurophysiology* **70**, 781–802 (1993).
5. Williams, S. R. & Mitchell, S. J. Direct measurement of somatic voltage clamp errors in central neurons. *Nature Neuroscience* **11**, 790–798 (2008).
6. Romand, S., Wang, Y., Toledo-Rodriguez, M. & Markram, H. Morphological Development of Thick-Tufted Layer V Pyramidal Cells in the Rat Somatosensory Cortex. *Frontiers in Neuroanatomy* **5**, (2011).
7. Cadwell, C. R. *et al.* Multimodal profiling of single-cell morphology, electrophysiology, and gene expression using Patch-seq. *Nat. Protoc.* **12**, 2531–2553 (2017).

REVIEWERS' COMMENTS:

Reviewer #4 (Remarks to the Author):

The authors have made considerable efforts to address my concerns. I still find the combining of a non-supervised classifier and an experimental finding of the matching of MC vs NonMC compared to Pyramidal vs stellate a mixed message but both aspects of the paper have value (the former I fear will end up being under-appreciated). Nonetheless both have their strengths and together certainly amount to something that is more than worth publishing in Nature Communications.

Reviewer #5 (Remarks to the Author):

Review of the re-revision of Scala et al. "Neocortical layer 4 in adult mouse differs in major cell types 1 and circuit organization between primary sensory areas"

I think the authors in general have done a good job to implement in a reasonable manner all the relevant criticism. I am pleased about the "reunification" of FS cells into a basket cell super-group. It would be helpful, however, if the authors could provide a/their definition of double bouquet (since I stated previously that this has never been done despite the wide usage of the term).

I also like the "functional implications" paragraph. However, I think that "25 μ m texture discrimination" should be improved to "25 μ m-spaced particle texture discrimination" etc., and that reference 74 (which is kind of redundant to 75) should be replaced by a classic in the field: Carvell and Simons 1990 (J Neurosci).

I found quite some typos or linguistic errors, which I find too tedious to list them here, except Methods, l. 527 where it has to be neuromorpho.org. "bitufted" is not separated by hyphen (bi-tufted). Please check the rest of the text carefully.

Specifically, however, there remains the "space clamp" issue, or, with other words, how serious any kind of connection probability can be taken, considering all the technical imperfections inherent to the methodology. I agree with the authors that cesium is not the solution but I still would claim that it is an important complementary approach. For instance, the reported zero connectivity between pyramids in V1 might discourage any further research on this whereas a few percent connection probability (like in S1) would make a qualitative difference potentially inspiring more research on this topic.

Anyway, it remains an ambiguous issue and I will not force the authors to do these experiments but the editor might think it over.

REVIEWERS' COMMENTS:

Reviewer #4 (Remarks to the Author):

The authors have made considerable efforts to address my concerns. I still find the combining of a non-supervised classifier and an experimental finding of the matching of MC vs NonMC compared to Pyramidal vs stellate a mixed message but both aspects of the paper have value (the former I fear will end up being under-appreciated). Nonetheless both have their strengths and together certainly amount to something that is more than worth publishing in Nature Communications.

We thank the Reviewer for this assessment. Splitting the paper in two separate parts would be very difficult, so we decided to keep it together.

Reviewer #5 (Remarks to the Author):

Review of the re-revision of Scala et al. "Neocortical layer 4 in adult mouse differs in major cell types 1 and circuit organization between primary sensory areas"

I think the authors in general have done a good job to implement in a reasonable manner all the relevant criticism. I am pleased about the "reunification" of FS cells into a basket cell super-group. It would be helpful, however, if the authors could provide a/their definition of double bouquet (since I stated previously that this has never been done despite the wide usage of the term).

We have inserted a definition.

I also like the "functional implications" paragraph. However, I think that "25 μ m texture discrimination" should be improved to "25 μ m-spaced particle texture discrimination" etc., and that reference 74 (which is kind of redundant to 75)

should be replaced by a classic in the field: Carvell and Simons 1990 (J Neurosci).

We thank the Reviewer for this suggestion. We have adjusted the wording and replaced the citation.

I found quite some typos or linguistic errors, which I find too tedious to list them here, except Methods, l. 527 where it has to be neuromorpho.org. "bitufted" is not separated by hyphen (bi-tufted). Please check the rest of the text carefully.

We have fixed these and other typos.

Specifically, however, there remains the "space clamp" issue, or, with other words, how serious any kind of connection probability can be taken, considering all the technical imperfections inherent to the methodology. I agree with the authors that cesium is not the solution but I still would claim that it is an important complementary approach. For instance, the reported zero connectivity between pyramids in V1 might discourage any further research on this whereas a few percent connection probability (like in S1) would make a qualitative difference potentially inspiring more research on this topic.

Anyway, it remains an ambiguous issue and I will not force the authors to do these experiments but the editor might think it over.

We thank the Reviewer for this discussion. We adjusted our formulations in the Discussion to make clear that further work is necessary.